# Citizenship Educational Policy: A Case of Russophone Minority in Estonia

**Nikolai Kunitsõn *** and **Leif Kalev**

School of Governance, Law and Society, Tallinn University, 10120 Tallinn, Estonia; leif.kalev@tlu.ee
* Correspondence: nikolai.kunitson@tlu.ee

**Abstract:** In the contemporary era, societies are divided, and political polarization is increasing. One of the most powerful instruments the government can use is general standard education, specifically citizenship education. We will look at the case of Estonia, because Estonia's main political cleavage is the ethnic cleavage between the Estonian and the Russophone community. Our main research question is as follows: How would it be possible to use democratic citizenship education to decrease in the future the socio-economic inequality between different communities in Estonia? We will outline the context of ethnic socio-economic inequality in Estonia and show how these differences have been at least partially influenced by the current education system in Estonia and how citizenship education can be used to reduce these inequalities in the future. We will conduct an empirical analysis of the curriculum, and this will be followed by semi-structured qualitative interviews. In the discussion, we will make suggestions to the current Estonian citizenship education policy and offer various insights into tackling this issue.

**Keywords:** citizenship education; inequality; minority education; democratic citizenship

## 1. Introduction

When Estonia regained its independence in 1991, it inherited a rather segmented society where the Russophone minorities made up more than one-third of its population. During the following decades, various policies have been implemented to improve the integration of different communities in Estonia. Although there have been some improvements (e.g., increased proficiency in the state language), the socio-economic status of the Russophone minority is still considerably lower. In addition, the Estonian de facto bilingual education system, which separates those two communities from an early age, raises the question: Can the Estonian education system offer equal opportunities for young people from different ethnic backgrounds?

On a broader scale, in the contemporary era of heterogeneous lifestyles and increasing political polarization, having a common societal culture and basic political coherence is increasingly a challenge for the whole society as political community. Citizens have grown distrustful of politicians and of the democratic institutions and process in general (e.g., Dalton 2004; Hay 2007; Papadopoulos 2013). As Hay (2007, p. 11) has noted, "Our sense of political citizenship in national democracies appears to be under threat."

One of the most powerful instruments the governments can use to balance these trends is the general standard education, through which a common societal frame of reference is developed. A democratic state needs conscious citizens with knowledge, skills, and attitudes. These are mostly generated via the education system, which usually includes some kind of civic and citizenship education, whether it be a separate course, a cross-curricular topic, or something else (e.g., Crick 1998; Stoker et al. 2012; Stoker [2006] 2016). This kind of democratic citizenship is vital for common identity in multi-cultural societies.

At the same time, these abstract ideas need to be implemented in concrete contexts, which can not ignore the realities of inequalities of different groups in society. Schools are

seen as a place wherein students from different socio-economic backgrounds could have a sense of common democratic citizenship, which is crucial for democratic citizens in the state. Empowering minorities goes beyond language skills and personal contacts; people also need competences to act as proactive societal, political, and economic citizens in a democracy. Those who lack those knowledge, skills, and values are in danger of further marginalization, including socio-economic aspects.

In this article, we will study the case of Estonia, analyze the citizenship educational policy with a special focus on democratic citizenship in the case of minorities. We focus on the civics and citizenship education course (hereafter civics) in Estonian secondary school with a key interest in selected aspects of public policy implementation and design. Estonia serves as a good case for the study: it has relatively developed framework curricula and syllabi and a separate civics subject. We study the content of civics and its reflection and implementation by the teachers.

Our main research question is, how would it be possible to use democratic citizenship education to decrease in the future the socio-economic inequality between different communities in Estonia? More specifically, we will explore, what are the suggestions of teachers to improve democratic citizenship education policy? First, we will outline the context of ethnic socio-economic inequality in Estonia by focusing on the demographic change that occurred in the 20th century, including the differences in language skills, education system, and regional differences. We will show how these differences have been at least partially influenced by the current education system in Estonia and how citizenship education can be used to reduce these inequalities in the future. More specifically, we will focus on the factors that are mentioned by the civics teachers in teaching children from minority groups.

Our approach is based on the concept of integration. We will not discuss normative proposals on arranging the Estonian society, but we will focus on democratic citizenship education based on the existing regulation with a special interest in educating minority students, based on the civics course, with the aim to understand the practical challenges and possibilities to improve the implementation of policies. We are not proposing an assimilation approach toward integration. We are broadly based on John Berry's (1997) acculturation model, which differentiates four approaches toward integration differentiated, depending on both the majority and minority group. This approach can be seen in Figure 1.

**Cultural Adaptation (relationship sought among groups)**

| | | Low | High |
|---|---|---|---|
| | **High** | Separation | Integration |
| **Maintenance of heritage culture** | **Low** | Marginalization | Assimilation |

**Figure 1.** Berry (1997).

This model is based on two dimensions: whether the immigrants find it important to maintain their ethnic culture and wherever they find it important to adopt the mainstream culture. In this article, we are suggesting that some kind of common societal frame is needed, which is provided by the democratic citizenship education in a formal education system.

This is especially relevant because in Estonia, the children from the biggest minority group—Russophone minority—often have the beginning of their education in the Russian

language and they learn in separate schools. However, using education as a programming instrument is not simple. The focal point is the role of teachers, since they are in the key position as front-line bureaucrats (Lipsky [1980] 2010). Their role is crucial both in designing the real-life educational content and integrating the top–down frameworks and pupils' everyday feedback into personalized practical strategies. In addition, we use Taylor's theory of curricula and Dewey-based pragmatist education to elaborate on selected aspects of policy design and implementation. We will illustrate our discussion with examples from curricula and from interviews with civics teachers with experience in schools with the minority population students.

We will start by explaining the context of Estonia and its socio-economic inequalities, with a special focus on its school system, which are followed by relevant concepts of citizenship education and Lipsky's front-line bureaucracy theory. This is followed by Taylor's curriculum theory and Dewey's practices of learning. Then we develop the methodology and conduct an empirical analysis of the relevant parts of the national framework curricula and subject outlines, followed by interviews with civic teachers from Estonian–Russian schools, with a special focus on teacher perspectives on their agency in qualitative interviews. Then, the results are elaborated in the Discussion and Conclusion sections.

## 2. Contextual and Theoretical Background

### 2.1. Estonian Society and Education System

Estonia is a small North European state with a population of 1.3 million people, which has experienced a significant demographic change in the demographic during the 20th century. In the second half of the 20th century, during the Soviet occupation period, Estonia experienced a change from a mono-ethnic state, where more than 95% of the population were ethnic Estonians, to a multi-cultural society, where the share of minorities rose up to 39% in 1989 (Tammaru and Kulu 2013). This largely Russophone minority is mostly characterized by their native language, which is Russian, and they constitute roughly one-third of the population of Estonia.

The minority issue has been one of the main cleavages in Estonian society and politics (e.g., Vetik 2012, 2015; Saarts 2017). This social, economic, and political problem has been addressed by the government by different measures, including neglecting, assimilation, and integration strategies, and one cannot say that there has not been progress (for example, the obtaining of Estonian citizenship, increase in proficiency in the Estonian language, etc.), but there are still a number of unresolved challenges. For example, in the capital Tallinn, more than one-third of the population consists of the Russophone minority; also, in one of the eastern counties in Estonia, called Ida-Virumaa, the Russophone community makes up more than 80% of the population. This makes the integration policies, also the school systems reform quite challenging. Around 86,000 people have Russian citizenship, and around 78,000 have no citizenship at all. In addition, recent migration statistics from the last decade show that a significant part of immigrants have either Ukrainian or Russian citizenship, which shows that this issue will be prominent in the near future (Estonian Statistics 2021).

Even though the Russophone community is not intrinsically homogeneous, in general, they have a notable disadvantage compared to Estonians in their socio-economic status (see further Soosaar et al. 2017; Pohla et al. 2016). For example, on average, the yearly disposable income shows that Estonians with Estonian citizenship earn more than 20% more than non-Estonians. In addition, non-Estonians with Estonian citizenship earn about 10% more than non-Estonians without citizenship (see Table 1, Estonian Statistics 2021) In addition, their willingness to acquire Estonian citizenship is decreasing, and trust for state institutions is lower in comparison to Estonians (Kaldur 2017). In addition, the Russophone community is under-represented for example in the national governmental sector (Ivanov 2015). In addition, the communities are separated in media consumption, geographical location, and crucially, in the education system.

**Table 1.** Equalized yearly disposable income by ethnic nationality and citizenship in euros (Estonian Statistics 2021).

|  | **2017** | **2018** | **2019** |
|---|---|---|---|
| Estonians with Estonian citizenship | 12,330.18 | 13,434.77 | 14,304.91 |
| Non-Estonians total | 10,300.19 | 11,379.71 | 12,212.64 |
| Non-Estonian with Estonian citizenship | 10,826.41 | 12,054.99 | 12,792.90 |
| Non-Estonians with other citizenship | 9777.56 | 10,704.88 | 11,629.19 |

As a legacy result of Soviet Union policies and demographic changes that occurred in Estonia, the education system has what is now called a "bilingual education system". The state-funded school system is in Estonian, but there are also state-funded schools that also use the Russian language in a significant amount, mostly for Russophone community children (Skerrett 2013). About every fifth student is studying in an Estonian–Russian comprehensive school (Põder et al. 2017). The differences in educational outcomes by PISA (OECD's programme for International Student Assessment) testing show a one-year gap between these communities in different schools (Põder et al. 2017; Täht et al. 2018). On one hand, this reflects the segregation and separation of these two communities, and on the other hand, it reinforces it (Hogan-Brun et al. 2008), having direct implications on the possibilities of integration and crucially, to the future socio-economic lower status of the Russophone community.

To be clear, Estonia has, strictly speaking, one school system with the same curricula, but there are different languages of instruction. The Estonian school system is based on four levels: pre-school, basic, secondary, and higher education. The basic compulsory education system is a nine-year comprehensive school. At this level, it is possible to study either in Estonian or in Russian. At the secondary level, it is possible either to study in Estonian or in the model 60/40, whereas 60% of the courses have to be taught in Estonian, including civics courses. These schools are mostly located in the capital Tallinn or in the northern–eastern part of Estonia, where the Russophone community is a majority in the cities such as Narva, Sillamäe, or Kohtla-Järve. Formerly, the instruction in these schools was entirely in the Russian language. The different school communities do not interact with each other during the compulsory education period, meaning that their social networks will differ significantly, which has a direct influence on the future labor market perspectives. Lindemann (2013) shows that the existing school system reproduces inequalities in the future; for example, the students from Estonian-Russian schools are less likely to continue their education in the higher education system (Lindemann and Saar 2011). In addition, data show that the results of students in Estonian–Russian schools are considerably worse than in Estonian schools. PISA (Programme for International Student Assessment) results show that by the age of 15, the gap between Estonian and Russian schools is 39 points, which equals approximately one year of education (Põder et al. 2017). These results are also evident in previous studies—2006, 2009, 2012, and 2015 (Täht et al. 2018, p. 5). Usually, these differences are explained by the language barrier, but the case of Estonia is exceptional in comparison to many other countries with minorities. There are two key distinctions. First, minorities in Estonia study mostly or partly in their own native language, meaning that the quality of their education is not based on the proficiency of state language, which is often the case for lower educational achievements. The second issue is related to larger socio-economic inequality. As mentioned above, the socio-economic cleavage between the minorities and Estonians exists in Estonia, but it is smaller than in other Western European countries (Lindemann 2013; Vetik and Helemäe 2011). In addition, PISA results show that there are no major differences in school background and organizational settings.

An ICCS (International Civic and Citizenship Education Study) 2009 study (Toots 2011) showed that interest in political and societal topics is higher amongst the schools with Russian language instruction, but in Estonian schools, students rate their knowledge of politics higher than their peers from Estonian–Russian schools. In addition, the same

study showed that minority students' knowledge about democracy, rule of law, civil rights, etc. is poor compared to their peers in Estonian schools (ibid, p. 35). A similar divide occurs from the perspective of teachers. The biggest difference is in teachers' approach of developing knowledge about social, political, and public institutions in the civics class. In Estonian schools, 49% of teachers thought it is important, but in Estonian–Russian schools, the figure was only 11% (ibid, p. 40). In contrast, teachers from Estonian–Russian schools seem to emphasize critical learning skills more than their Estonian peers. A more recent ICCS study (Toots and Idnurm 2018) shows that children who study in Russian schools know less about society and their civic participation is also lower and there are two main reasons for this. They claim that the Russian children are still looking for their role as citizens in Estonian society and that institutions are not so important for young people, and they conclude that value-based state identity strengthening could be an important part of active citizen participation.

In addition, as we will demonstrate later, the distinguishing of knowledge, skills, and values is crucial. In democratic citizenship education, more focus is needed on the aspect of practical skills, which is more complicated in the Estonian–Russian schools. There are three aspects to this issue. The first is related to the weaker state identity of minority students, and second is their relative lower socio-economic status, which means that their background knowledge of civics is already weaker, which means teachers have an even more salient role in this educational process. The third is that enhancing the practical skills and readiness to take an active role as a citizen needs more support and experience for the more detached students.

To sum up, the bilingual education system in Estonia fails to address the issues of minorities from two different aspects. First, students still struggle to achieve sufficient proficiency in the Estonian language, which has clear implications for their future studies and also for their future socio-economic status in general. Secondly, since the communities are separated also in terms of their media consumption, their common democratic citizenship is weaker in comparison to Estonians (Kaldur et al. 2017), which presents future problems in terms of their education and future as democratic citizens in Estonia. To be clear, it is not a problem in itself if people follow different media. The problem is if they do not follow Estonian media at all, as it is in some cases in practice (people follow the media of the Russian Federation). This means that the people even lack key information, not to mention sufficient knowledge of the host society, which in turn means that they lack the basis to act as democratic citizens in a state.

One of the solutions to the above-mentioned issue could be an institutional reform, which means implementing a "common" or "unitary school" system, where children with different native languages would study together. This could tackle the above-mentioned two problems: (1) students from minority backgrounds would achieve sufficient Estonian language skills, which would help them in the future in the labor market and therefore reduce the socio-economic inequalities; and (2) it would increase the democratic citizenship of minorities, which is a necessary component in order to be a successful citizen in Estonia. This model has not yet been implemented because of different political preferences and strategies, and there has been somewhat strong resistance from the Russian minority. However, recent studies (Kaldur et al. 2017) show that the majority of the Russian-speaking minority is now favoring this model and there is a rhetorical consensus in Estonian political parties, considering this issue.

### 2.2. Democratic Citizenship Education

Broadly, the good democratic citizen is a political agent who takes part regularly in politics locally and nationally, not just on primary and election day. Active citizens keep informed and speak out against public measures that they regard as unjust, unwise, or too expensive. They also openly support politics that they regard as just and prudent. Although they do not refrain from pursuing their own and their reference group's interests, they try to weigh the claims of other people impartially and listen to their arguments. They

are public meeting-goers and joiners of voluntary organizations who discuss and deliberate with others about the politics that will affect them all, and who serve their country not only as taxpayers and occasional soldiers but by having a considered notion of the public good that they genuinely take to the heart. The good citizen is a patriot (Shklar 1991, p. 5).

However, on a slightly deeper look, citizenship is a manifold concept (e.g., Heater 2004; Guillaume and Huysmans 2013; Shachar et al. 2017). This is further amplified in citizenship education where different approaches, aspects, and normative perspectives are complemented with the organizational and pedagogical considerations: whom to educate, via which structures, in which ways, with which aims, etc. (see e.g., Reid and Gill 2013).

Citizenship education can be a powerful tool for preparing and developing citizens for political and also broader societal and economic life (e.g., Crick 1998; Stoker et al. 2012; Westheimer 2015; Stoker [2006] 2016). Thus, it is potentially one of the key areas of education for empowering minority students.

Citizenship education can be studied as a focused course of civics or as the outcome of various courses that support some social and political competencies. An even broader perspective would follow a Dewey (1910) understanding of all education as integrating citizens and society through an approach based on democratic values and practices. In this paper, we focus on citizenship education as regulated by the government and conducted in general schools, i.e., civics. This allows studying how the public authorities seek to steer the preparation of citizens, and how it is implemented and experienced in practice.

From this perspective, citizenship education can be analyzed as policy design and implementation, combining top–down and bottom–up perspectives and emphasizing meaning-making (e.g., Jennings 1996; Spillane 2004; Lester et al. 2017). We start from the national framework curriculum for secondary school (Estonian Government 2011b) that establishes the general aims, objectives, competencies, and other items every school is expected to follow. Then, we study how these are further developed in the national curriculum of social studies and the implementation practices of civics teachers. Then, we move to the experiences and reflections of educational practice by teachers who teach at schools with minorities.

Teachers can be seen as street-level or front-line bureaucrats who engage people directly and shape the practice of implementation. The policy-making roles of street-level bureaucrats are based on two interrelated aspects in their positions: relatively high degrees of discretion and relative autonomy from organizational authority. The position of street-level bureaucrats regularly permits them to make policy with respect to significant aspects of their interactions with citizens (Lipsky [1980] 2010, p. 13).

Street-level bureaucrats exercise wide discretion in decisions about citizens with whom they interact. This is also the case in teachers' decisions on the content of teaching. The individual actions of street-level bureaucrats are an important part of agency behavior. The discretion arises of the character of their professional activity that calls for human judgment that cannot be fully programmed and for what machines cannot substitute (Lipsky [1980] 2010, p. 161). Here, sensemaking is important in understanding how educators think about the implementation of policy and more broadly understand their work (Spillane 2004; Hogan et al. 2018; Tan 2017), including one's own role and strategy and how it relates to both the top–down and bottom–up contexts.

Partly opposite to this, Scott and Lawson (2002) express reservations about the efficiency of schools and teachers. They claim that a curriculum is not a neutral document. Any statement of what is to be learned is permeated with objectives and intentions. If learning outcomes are closely defined, it is both possible and likely that the achievement of those outcomes will be assessed and quantified. Manifest and palpable assessment goals—learning by objectives—inevitably lead to learning and teaching patterns that are dominated by the requirement to meet the defined and measurable objectives, leaving aside the ones that are difficult to measure. This is more powerfully expressed by Westheimer (2015, p. 19): due to standardization, "no teacher is left teaching".

We will not approach teaching as entirely top–down, mechanical, and performance measured, nor as developed entirely bottom–up by the street-level bureaucrats. This also means a middle way in terms of implementation studies. Traditional implementation studies have typically looked at whether bureaucrats' practices align with formulated policy goals (Hupe et al. 2015), or how lower-order bureaucrats carry out orders of higher-order principals (e.g., Brehm and Gates 1997). In the newer studies, researchers have turned their attention toward the practices influenced by policy as well as the ways in which these practices influence policy development (Raaphorst 2019). Our key focus will be on teachers, their understanding, and contexts, but we will discuss this in the broader picture of civics as a national policy resource.

The key aspects in citizenship education are knowledge, skills, and attitudes (Heater 2004, p. 343). Knowledge is related to facts, interpretation, and personal roles. Attitudes are related to self-understanding, respect for others, and values. Skills are related to intellect and judgment, communication, and action. This distinction is salient: a democratic citizenship policy should contain all three elements because only then do they complement each other.

### 2.3. The Practices of Curricula and Learning

In analyzing the Estonian civic education system, it is important to see how the curriculum is developed and how the learning process is happening. We will use an approach for curriculum analysis broadly based on Tyler (1949) and the perspective on the learning process broadly based on Dewey (1910). We see these as mutually complementing, not opposite.

Tyler (1949) provides a linear rationale for viewing, analyzing, and interpreting the curriculum, which is one of the most widespread ways to view curriculum as an instrument of education. His model consists of four basic principles: (1) defining the learning objectives, (2) establishing useful learning experiences, followed by (3) organizing learning experiences, and (4) evaluating the curriculum. The first step is considered to be the most important one, or as Tyler (1949, p. 3) has put it, "If we are to study an educational program systematically and intelligently, we must first be sure as to the educational objectives aimed at." Tyler sees education as experience; he approaches this from the perspective of the problem-solving process and sees assessments as evaluations rather than measurements.

Similarly, Dewey (1910) discusses education as the process of changing the habits of people. Dewey has stressed that when teachers teach—no matter if it is "good" or "bad"—students are still developing habits. The habits that children have acquired from their previous experiences work fine until they are encountered with problem-situations, something where their previous habits do not work anymore. This is where the learning takes place—when students encounter problem-situations, they reflect upon these, identify different possible solutions, and ideally find a contextual resolution to the case.

The aim of a problem-situation is to "generalize", but generalization should not be an end, rather as a means to better deal with future problems. The key idea of the Deweyan model of education is that the role of the teacher should be creating a problem-solving mindset in students.

We use the model of Taylor to elaborate on the Estonian curricula and civics course outline and Dewey educational practices in explaining the role of skills in democratic citizenship education.

## 3. Methods

The empirical research consisted of two parts. In the first part, the text of the high school framework curriculum and social subjects outline was analyzed via content analysis. In the second, semi-structured qualitative interviews with teachers from Estonians and Estonian–Russian schools were conducted, and they were analyzed also via content analysis. Our results are presented in three categories: curriculum analysis, teacher inter-

pretation, and discussion following democratic citizenship education in Estonian–Russian schools.

As stated above, our main research question is as follows: how would it be possible to use democratic citizenship education to decrease in the future the socio-economic inequality between different communities in Estonia? We have shown how inequality in socio-economic spheres is connected in the case of Estonia at least partly with democratic citizenship education and identify the challenges brought on a separated education system. Our empirical analysis research questions are the following:

1.  What are the main findings in Estonian high school curricula and civics course subject outline? Are they comprehensible?
2.  How is civic education presented in Estonian high school curricula and a civics course subject outline?
3.  How do the civics teachers transform the regulations in high school curricula into the content of their subject? What is the role of the teacher agency in this process?
4.  What are the specific challenges of teachers in relation to the students from the Russian minority in Estonia?

In Estonia, the core document for the content of education in general schools is the national framework curriculum that states both general aims, values, cross-curricular topics, etc. as well as the subject-related objectives, topics, etc. There is one national curriculum for basic schools (Estonian Government 2011a) (grades 1–9) and another for secondary school (grades 10–12); we focus on the latter. We study the curriculum on two levels. First, we analyze the general values, competencies, and cross-curricular themes that are in principle binding for all the subjects. Second, we will take a closer look at how this is elaborated in the civics subject outline.

As a next step, the semi-structured qualitative interviews were conducted with the civics teachers in secondary school who teach in Estonian–Russian schools or in schools with a majority of Russian background students. First, a draft framework for the interview was developed. This was based on our previous analysis of the national framework curriculum and civics subject outline. We had an interview frame, which was used based on the context (available in Estonian) and if needed, we added some question to discuss issues in-depth.

Then, the interview was piloted in an interview with a civics teacher and further developed based on the reflection of this experience. This was done via the Internet due to the COVID-19 situation. We followed the frame and elaborated upon the aspects in case of shallow answers but also encouraged the teachers to discuss the aspects they recognized as important to understand teachers' subjective perspectives and rationalization strategies as well as to further enhance our understanding of the practical educational context. We proceeded with the interviews up to the saturation of the sample, in a total of 10 interviews. We chose the people based on a set of predefined values: they had to have an experience in teaching civics class in Estonian–Russian language schools. We contacted teachers via various sources and used a snowball method to find other participants.

Saturation was assessed based on the new perspectives the interviewees presented on the teacher's role and teaching strategy. Before the interview, all the teachers received outlines of high school curricula and of civics subject themes. The use of the materials during the interviews was voluntary and teachers used them in different amounts. As the autonomy of teachers is significant, we approached them as experts.

We used content analysis, based on our research interests. Both authors analyzed the interviews separately, and later, we discussed the findings and then deliberated and synchronized the main findings, similarly to the curricula analysis.

## 4. Results

### 4.1. Curriculum Analysis

The framework curriculum starts with the underlying values, which are differentiated into two groups—humanistic (honesty, justice, etc.) and societal (freedom, democracy, etc.).

The curriculum goes on to outline the "culture and value competencies" (the title of the subsection) that consist of general humanistic and moral claims. The competencies also stress active citizenship.

The general objective of social studies is stated as "developing students' social competence", with the emphasis on "understanding the causes and effects of the social changes". This is followed by "knowing and respecting human rights and democracy" and other similar objectives that point to the obligations and rights of the citizen but are not treating the citizen as an active agent in society. The first mention of active citizens is in the aims of civics. From four area topics and 14 topics that are most emphasized, only two are directly related to the role of citizen. Most topics are connected to the level of knowledge, and in particular, they tackle the issues of society, politics, economy, and international relations. This means that on the level of citizenship, the citizens are mostly portrayed as rather "consumers" or passive subjects, not carriers of active democratic citizenship. To conclude, it can be said that the general aims of social studies relate vaguely to democratic citizenship.

The learning outcomes of social subjects are more specific. For example, it is written that the student must "know and value the principle of democracy", and it is essential that he identifies himself in society, taking into consideration his possibilities, ability to manage in a market economy, etc. Most of the topics are addressing the system level, ranging from national and local political and legal systems to societal stratification, consumer behavior, domestic and global economy, international political system, and the operation of the European Union. However, as in the general framework curriculum, also here the role of the citizen is in the background.

The framework syllabus (subject outline) of civics subject consists of two courses—"Governance of democratic society and citizen participation" and "Economy and world politics". The first one is of key interest here, as it covers the political, social, and legal topics.

The subject outline mostly focuses on introducing different institutions but lacks the connections of citizens to institutions. There is a significant mismatch between the course description and the learning outcomes that emphasized personal development as a citizen. In the subject outline, the focus is on knowledge. Skills and attitudes are mentioned only briefly. There is a missing link to the citizen being an agent, having an active role. The second "civics" course is not directly focused on the role of the citizen, encompassing the general operation of the economy and the international system.

In sum, the national documents contain elements of democratic citizenship but do not present a balanced and systematic strategy remaining eclectic and partly controversial. For example, if we take the well-known divide of normative perspective to citizenship (liberal democratic, civic republican, and national communitarian) (see Delanty 2000; Lister and Pia 2008), then all the citizenship normatives are represented, but the active citizen, which is mostly related to civic republican normative, is the least mentioned one. Our result show that it does not present a balanced a systematic strategy from the state, meaning it is eclectic and partly controversial.

The emphasis on knowledge leaves skills and values little addressed. We interpret it as a problematic point. This shows that on the declarative level, the democratic citizenship is addressed, but it is unclear what kind of "good citizens" the formal education system should shape. Thus, the role of the teacher as the practical designer and implementer of the course becomes substantive. However, the current teacher training curricula provide at best a limited basis for this (Jakobson et al. 2019). In addition, as we have shown previously, the students in Estonian–Russian schools are already lagging behind in this field, which means that without a clear aim and implementation policy, they might be even more marginalized in the future.

## 4.2. Teachers' Perspective

In this subsection, we first examine the understanding of teachers about citizenship education and therefore the role of the citizen. We also address the issue of national curricula. This is followed by the peculiarities of the Estonian–Russian school system in Estonia. The outlines from interviews are translated by the authors.

The interviewed civics teachers are broadly familiar with the secondary school curricula as could be expected. They have a general knowledge of curriculum aims, knowledge, attitudes, and skills. The curriculum is considered to be a fundamental and basic part of the teaching framework but is not taken as a strict guideline, rather as a broad guidance and a large pool of objectives and topics, which is used by teachers to "cherry-pick" the knowledge, values, and skills they wish to address. They did not find that any of the curriculum aspects are irrelevant and mostly agreed with these, but some added further humanistic values there (e.g., honesty, creativity, personal integrity).

> **Respondent:** They are like, fantasy, in some ways, fantasy, maybe, it is like this idealistic approach.

> **Respondent:** In high school, they are very abstract, that the student is a responsible citizen, who knows the main principles of how the society works and this is what I have formulated the most for myself.

Teachers stressed that their job is to influence and prepare the students for adult life in society. This emphasis fits well with the idea of street-level bureaucracy, where the self-aware bureaucrats see themselves as confident and influential actors, e.g., as "activist teachers" (Leonard and Roberts 2016).

> **Respondent:** If we talk about competencies, then you need to stress that the student would be aware of what democracy is and why we need it.

> **Respondent:** Well, hmm, I think, it is important that the students know how the state is operating. And another thing, that all citizens should know, is their basic rights and responsibilities.

> **Respondent:** I personally educate a simple person, who feels himself needed in the society, who feels that he can achieve a change in society, who knows his rights, and well, that he is a fully valued member of society.

Teachers reflected on their active role in the civics course. They saw that it was a two-sided role—on one hand, they had the freedom to design the course, but at the same time, it was time-consuming and therefore also depends a lot on the teacher's agency.

> **Respondent:** And every teacher like, he swims in the sea, he doesn't have a boat, he needs to build it himself, and I tell you, I tell you honestly, this is very hard, that I, as a teacher, have to develop all these materials, first of all, it is tiresome, second, well, maybe I'm not competent enough for this.

> **Respondent:** Estonian teachers are actually very happy, she is very free in her decisions, she chooses her own method and the way she does things, but there is the thing that for active learning methods, the curriculum is too much.

Teachers mostly discussed knowledge, while the attitudes and practices were not reflected in the same depth. As a result, democratic citizenship is largely unattained, or to be more precise, it is mostly addressed on the level of knowledge. However, to an extent, this is balanced by the personalized design of the course. As mentioned before, teachers' agency plays a key role in the teaching process. All the interviewees stressed their role of the "banking model" education, where they are the holders of information and transfer it to students. For this, they used their personal experience and practical examples to engage students meaningfully. Teachers did not believe that they had a significant influence on students' values and skills; rather, they hoped that by focusing on knowledge, the values and skills will follow.

> **Respondent:** Oh, that state tries to achieve via curriculum, I'll tell you honestly, that this curriculum doesn't do anything without a teacher, teachers always can shape the student like he wants and curriculum doesn't stop it...

> **Respondent:** Hmm, I think, they are enough to achieve, but at the same time, my experience as a teacher shows that they can't be always achieved. They are in some ways, like fantasy, it is this kind of idealistic approach.

While teachers are familiar with the curriculum, their personal reflection of their role and personalized teaching strategy (consciously or unconsciously) influenced the knowledge, attitudes, and skills they aimed to "transfer" to students. This reflects Dewey's idea of the role of the teachers' habits in the process of education and aligns our study with previous findings that teacher agency plays a salient role (Hogan et al. 2018; Tan 2017).

The teachers emphasized in-class interactive methods, such as group work, debates, and other practical exercises. Those methods were used to keep the students interested, to make the class more interesting. Most of them did not address methods as a possibility to influence the values and skills of students. So, in general, the practical assignments provided students with even more knowledge but no real practice in being a "citizen". Teachers were not overly optimistic about the perspective of students obtaining practical experience outside the classroom. For example, the connection to local government, politics, and communities seemed much to depend on the personal connection of teachers with these institutions.

> **Respondent:** By using these active teaching methods, you actually can develop and bring forward those values and skills.

> **Respondent:** If Narva wouldn't be that far, from the center, where all these meetings take place. No, we don't do these kinds of simulations.

> **Respondent:** Yes, I have heard about these methods, but I don't use them. Well, yes, I know, that people use them.

Teachers also discussed the overload of the curriculum. The course has a vast number of topics but not nearly enough time to cover all of them. The topics range from the basic knowledge about the state and institutions up to the economy, European Union, and international politics. Teachers reflected on the need to make optimizing choices.

> **Respondent:** But the question is at the expense of what? I know that 20 min is not enough for discussion, there are different questions and then a question arises—should I deal with the structure of parliament or I forget about it and do it next time? Okay, that is possible, but in the next class, there is another topical issue, for example, the closing of EU borders? And again, the question arises, how?

> **Respondent:** And yes, if you want to do something practical, or some active learning method, it takes time and because of that, you can not introduce some basic principles of European social welfare.

### 4.3. Challenges with Democratic Citizenship Education in Estonian–Russian Schools

In comparison to Estonian students, teachers who taught Russian students mentioned several key issues that differentiate them from the teachers in Estonian schools. One of the key issues was the different media sphere where students and their families live, which is often the one of the Russian Federation. This is problematic from to aspects: first of all, teachers need to put in extra work to provide the basics contextual information and knowledge on Estonian society to the Estonian–Russian-school students in comparison to the students who study in Estonian schools, meaning that this leaves less time and possibilities to develop other aspects of democratic citizenship. Secondly, Russian state-sponsored media propaganda is often promoting anti-democratic tendencies, which also undermines principles of democratic citizenship.

**Respondent:** Oh yes, Estonian news—they do not read it. And of course, it depends, if the students are more focused on learning, they follow Estonian news. But if I look at the typical high school, then I need to force them to follow Estonian news.

**Respondent:** Teachers can't influence a lot; well, the children come from home, then the role of the teachers is to guide them and to offer alternatives to what is taught to them from home.

**Respondent:** Just recently was this simulation, but unfortunately, I didn't invite my students, my students are in that sense more passive, but maybe they wouldn't be so passive if Narva would not be so far from the center, where all these meetings are happening. So no, we don't do these simulation activities.

**Respondent:** Well, we have to fight very viciously with this Russian propaganda machine, and refute a lot of myths like this.

**Respondent:** It is largely the question of the information sphere and in what country they think they live. Well, if they need to draw a president, then some of them draw Lukashenka.

A related feature is the underlying cultural differences that influence not only attitudes but also self-esteem and learning habits. In case a student is less interested in society and politics and less disposed to work with skills and attitudes, it will be harder for the teacher to develop him or her toward more agency and engagement.

**Respondent:** I think it influences because my students are used to just memorizing things.

**Respondent:** Since I have a lot of students from Russian families, then, well, very often they try to explain to me that look, for Russians, everything is different from Estonians, we value totally different things.

**Respondent:** It also depends on the home environment, well, if the family is with lesser culture and societal interests...

Another main issue that teachers brought up was the separation of skills, knowledge, and practices. The majority of the teachers agreed with our finding in curriculum analysis, that while the role of active citizens was emphasized in general, the curriculum does not elaborate the practical objectives, strategies, examples, and possibilities to implement it.

**Respondent:** One thing is theory and another thing is practice. They can be full of theories, but if they don't know how does it exactly work, then the theory is not very useful.

**Respondent:** Well, basically, I think you might be correct, but another thing is how to implement this practical learning, because the school here is quite conservative, and if you have, like, this, outside school activities, then, to be honest, it is very difficult. It is easier in Estonian schools.

**Respondent:** But yes, in that sense, doing it yourself, there is not clear enough of this in schools. Everybody is in classes, and they do a small PowerPoint presentation.

On the positive side, it is worth mentioning that all the students who reach the 12th grade in Estonia usually speak already fluent state language, so the issue of language is not so salient. The larger question is that the students with weaker language skills do not enter high school.

**Respondent:** But still, sometimes, you don't explain some things, don't discuss some things, because of my language skills and because of their inability to express their feelings in a foreign language.

**Respondent:** Those people, who get more forward, there are so few of them, the Russian-speaking ones.

Several teachers supported the institutional reform of the education system, to reduce segmentation. However, this is not enough. For developing democratic citizenship, we need to focus more on aspects such as enhancing personal and civic competences and agency, balancing the influence of the Russian media sphere and the home environment. In addition, more emphasis is needed upon the methodology and language skills of teachers. This will be further elaborated in the discussion part.

**Respondent:** We need a common education system. It is as easy as that.

**Respondent:** Yes, earlier language teaching, and I would like it to be two-sided.

## 5. Discussion

Our discussion is divided among two main lines: first, we will discuss the findings from the analysis of curriculum and teachers' interviews. Second, we will examine these results in a more general overview of the Estonian civics education sphere.

Estonia has clearly structured and elaborated national high school general curriculum and civics course subject outline. One of its main aims is to develop active citizens, but unfortunately, it does not provide comprehensive logic on how to achieve it. This is a classic example of policy design with an open implementation—the general aim is clearly stated, but the implementation of it is left largely to the street-level bureaucrats, in this case, the teachers. However, it is not clear that this is entirely planned, as there is a relative mismatch between the objectives and subject outline and there is little training for the teachers in terms of the content of civics. High school teachers in Estonia have a relatively large autonomy, which on one hand, allows them to individually design the course, but at the same time, it can be very time-consuming and challenging.

Based on research, by implementing the Taylor curricula analysis, even though the general curriculum has stated its goals and aims, the curriculum is still mostly knowledge-based, leaving the aspect of skill and values in the background. This is not sufficient to teach democratic citizenship to the students. In addition, following the Deweyan pragmatic education model, more focus is needed to develop the curriculum and teaching toward more practical examples, such as group work, role-play exercises, etc. and practice in the community, in order to learn via practice, not just on a theoretical basis, because democratic citizenship is not only theoretical but also it consists of practices—acts of citizenship.

Our interviews with teachers showed that there are few main points that need to be clarified and addressed. We address the issue from two perspectives: general and Estonian–Russian school specific. From the general perspective, the curriculum is overloaded with different topics and themes, and while it seems they might be covered with the length of $2 \times 35$ h in high school, in order to truly grasp the active citizen concepts, it is not sufficient. In addition, the curricula and teachers stress the importance of knowledge, which is crucial, but the levels of skills and values are therefore left in the background. The teacher's agency, from the Lipsky street-line bureaucratic theory, is two-sided: it provides teachers with autonomy but requires a tremendous effort from teachers.

The analysis of the interviews with Estonian–Russian school teachers revealed that they have some specific issues that require further attention. Therefore, teachers need to make an extra effort with the Estonian–Russian school students in order to increase their democratic citizenship. We see the emancipatory power of democratic citizenship in the next aspects: First of all, because of the lower language skill, a home influence that comes from a lower socio-economic status, and the Russian state media influence, students have lower knowledge of Estonian society and democratic practices, which means that teachers need to emphasize the basic elements of the society even more. Secondly, teachers need more time and to put in extra effort in order to enhance the possibilities, knowledge, skills, and values in order for these students to participate in society as democratic citizens.

Our results show that even when teachers' autonomy is large, this alone is not sufficient to reach the aims of citizenship education in Estonia. Some additional tools and resources are needed, especially with regard to time and solutions for interaction and practicing. The current situation has its benefits, but it also can further increase the inequal-

ity in education, based on teachers' capacities and willingness to put in the extra work. More emphasis is needed toward methodological issues, such as how to develop practices of democratic citizenship, not only the knowledge. This could be achieved not only by changes in curricula and civics course subject outline but also in teacher training.

## 6. Conclusions

From a broader perspective, the Estonian separated school system reproduces the inequality not only in socio-economic or labor market aspects but also in terms of common democratic citizenship. The current solution in citizenship education remains rather thin, and thus, pre-existing differences in civic competences are not sufficiently mediated. Furthermore, it is doubtful as to whether it reaches its aims and objectives in Estonian–Russian schools or in Estonian schools, which means that not only democratic citizenship is not reached on the same level, but this also has negative effects in the future considering the larger socio-economic cleavage.

This study, among others, shows the potential solution to this issue by proposing the idea of the common or united education system in Estonia. However, the development of citizenship education is not only a matter of institutional reform. More emphasis is needed upon the critical skills to balance the Russian media influence and sometimes also the home environment. This is necessary in order to reduce marginalization and to develop democratic citizens, not just subjects. There are different options for this, for example on the institutional level of the curriculum design, but also on a more practical level, the methodology of teaching, such as interactive methods, for example, group work, discussion, simulations, etc.

**Author Contributions:** Both authors have participated in all of sections, except for conducting, transcribing and translation which were done by N.K. Both authors have read and agreed to the published version of the manuscript.

**Funding:** This research was partly funded by EU Horizon2020, grant number 857366, project MIRNet—twinning for excellence in migration and integration research and networking.

**Institutional Review Board Statement:** The study was conducted according to the guidelines of the ethics of Tallinn University.

**Informed Consent Statement:** Informed consent was obtained from all subjects involved in the study.

**Conflicts of Interest:** The authors declare no conflict of interest.

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
