# Peer review of "Citizenship Educational Policy: A Case of Russophone Minority in Estonia"

_socsci, doi:10.3390/socsci10040131_

Round 1

Reviewer 1 Report

Although this paper addresses a relevant topic, which is citizenship education as a means to promote the social integration of the Russophone minority in Estonia, I cannot recommend its publication. I list below the weaknesses that I detected and that explain my decision:

  • I have some concerns regarding the motivation of the study, which I understood is the integration of the Russophone minority in Estonia through citizenship education. The language used to frame this problem-solution is disconcerting because it seems that the author/s are seeking for an assimilation of the Estonian values and culture by this minority rather than their real inclusion, which would require questioning the dominant culture/values that persistently generate these social inequalities (and questioning the ICCS tests that justify them). This bias is also observable in the reading of the data when, for example, ‘not following Estonian news’ is interpreted as a sign of political disaffection.
  • The research questions are not clear. At the beginning of the article, it is said that the research questions are: ‘how would it be possible to use democratic citizenship education to decrease in the future the socio-economic inequality between different communities in Estonia? What are the suggestions of teachers to improve democratic citizenship education policy?’ (p. 2) But I don’t see how the data answers any of these questions. At the end of the article, where the methods are explained (which I don’t understand why this is located at the end), it is stated that the research questions were other completely different ones. This is very confusing and needs to be clarify.
  • In relation to the curriculum analysis, I feel that the author/s should delve much deeper into the data. The analysis could go beyond stating that the curriculum puts more emphasis on civic knowledge than on skills and attitudes. Within the knowledge dimension, for example, educational proposals that focus on the study of social problems and public controversies are not the same as the ones that focus on the study of civic institutions. However, the author/s do not mention anything about what particular knowledge is promoted in the curriculum. More refined conceptual categories are needed to analyze the data.
  • With regards to the presentation of the teachers’ perspectives, there is a mismatch between the author/s’ descriptions and the selected quotes. Often, the quotes do not reflect what is stated by the author/s. I would suggest including more quotes that evidence what is stated. I would also recommend to include longer quotes; some of the data looks very rich but is hard to appreciate in such short pieces.

I hope these suggestions would help the author/s to improve their paper. The topic is relevant, and the study has potential. Yet, further work needs to be done before publication.

Author Response

Dear Reviewer

Thank You for the helpful comments and constructive critique. We have added the revisions in the file using the „Track Changes“ function. We have corrected some typographical errors and made some minor adjustments, which are also highlighted. We address the issues raised in the following table.

Suggestions raised by the reviewer

Comment

Specific lines

I have some concerns regarding the motivation of the study, which I understood is the integration of the Russophone minority in Estonia through citizenship education. The language used to frame this problem-solution is disconcerting because it seems that the author/s are seeking for an assimilation of the Estonian values and culture by this minority rather than their real inclusion, which would require questioning the dominant culture/values that persistently generate these social inequalities (and questioning the ICCS tests that justify them). This bias is also observable in the reading of the data when, for example, ‘not following Estonian news’ is interpreted as a sign of political disaffection.

We have added a section that clarifies that our idea is not based on the assimilation ideas, but rather on integration - having some common identity in the state.

We added a section about the differentiation between integration and assimilation perspectives and believe that it is now more clear. In addition, we have clarified the inclusion aspect in other circumstances, e.g following the news.  We have added a section to clarify, why is “not following Estonian news” problematic in the case of Estonia.

  1. 68-97
  2. 243-247
  3. 576-585
  4. 676-679
  5. 682-691
  6. 777-778

The research questions are not clear. At the beginning of the article, it is said that the research questions are: ‘how would it be possible to use democratic citizenship education to decrease in the future the socio-economic inequality between different communities in Estonia? What are the suggestions of teachers to improve democratic citizenship education policy?’ (p. 2) But I don’t see how the data answers any of these questions. At the end of the article, where the methods are explained (which I don’t understand why this is located at the end), it is stated that the research questions were other completely different ones. This is very confusing and needs to be clarify.

We have clarified the main research question and more detailed questions for empirical research. 

We have a main research question and we have different research questions for empirical research. 

We structured the text, so it would be more logical.

  1. 57-59
  2. 375-379

In relation to the curriculum analysis, I feel that the author/s should delve much deeper into the data. The analysis could go beyond stating that the curriculum puts more emphasis on civic knowledge than on skills and attitudes. Within the knowledge dimension, for example, educational proposals that focus on the study of social problems and public controversies are not the same as the ones that focus on the study of civic institutions. However, the author/s do not mention anything about what particular knowledge is promoted in the curriculum. More refined conceptual categories are needed to analyze the data.

We have evolved more fruitful concepts and expanded our analysis to cover the normative aspects of citizenship education in paragraph 4.1 and in the discussion section.

  • 446-449
  • 456
  • 470-475
  • 649-651

With regards to the presentation of the teachers’ perspectives, there is a mismatch between the author/s’ descriptions and the selected quotes. Often, the quotes do not reflect what is stated by the author/s. I would suggest including more quotes that evidence what is stated. I would also recommend to include longer quotes; some of the data looks very rich but is hard to appreciate in such short pieces.

We have revised the teachers’ perspectives and made the requested changes. In addition, we have added some extra quotes that will exemplify our findings.

  • 504
  • 536-537
  • 546-547
  • 555-558
  • 576-585
  • 610-613628-632
  • 682-691
  • 694-696

Reviewer 2 Report

The topic approached by the authors is interesting and current, which can be published, after making major adjustments in the structure of the paper, for the paper to meet the requirements of the journal.
We recommend a restructuring of the paper in the standard format, with: introduction, theoretical background, Method, Results, discussions and conclusion. We consider that the presentation of some answers of the respondents is not opportune, not being followed by references on the authors of those included statements, but the content of this information can be interpreted and reproduced by the authors in their own presentation. To define the table 1, from definition, the web link can be moved to the list of final references, for the rigor of the citation.
The paper must be end with the section of conclusions, which now it isn't inside the paper. A numbering of the references would be useful for highlighting the sources consulted by the authors.

Author Response

Dear Reviewer

Thank You for the helpful comments and constructive critique. We have added the revisions in the file using the „Track Changes“ function. We have corrected some typographical errors and made some minor adjustments, which are also highlighted. We address the issues raised in the following table.

We recommend a restructuring of the paper in the standard format, with: introduction, theoretical background, Method, Results, discussions and conclusion.

We have made the requested adjustments.

We consider that the presentation of some answers of the respondents is not opportune, not being followed by references on the authors of those included statements, but the content of this information can be interpreted and reproduced by the authors in their own presentation.

We have elaborated on this aspect of presentations the answers and added some extra quotes. In addition, we have elaborated in the method section how we have selected the quotes and explain our analysis process

  • 404-428
  • 536-537
  • 576-584
  • 610-613

To define the table 1, from definition, the web link can be moved to the list of final references, for the rigor of the citation.

We have removed the web link in Table 1

The paper must be end with the section of conclusions, which now it isn't inside the paper

We have restructured the paper.

. A numbering of the references would be useful for highlighting the sources consulted by the authors

We have added the numbering to the references.

.

Round 2

Reviewer 1 Report

The authors have made a worthwhile effort to incorporate the suggested recommendations and have adequately and carefully argued the revisions. I do believe that the manuscript has considerably improved since the first version. In particular, the methodology and findings sections are presented in a much more robust way. In my opinion, the article deserves to be published in Social Sciences.

Author Response

Dear reviewer   We would like to thank You for the constructive critique and recommended suggestions. We believe that because of Your revisions our paper has improved. In the second round of reviews, we have made the following changes:
  • We have deleted a paragraph because it is repetitive (lines 672-680). This includes the reviews marked lines 678-680. This is elaborated now in lines 682-691.

  • We have separated the Discussion and Conclusion. The conclusion starts from line 701.

Reviewer 2 Report

The paper can be published with minor revision. The conclusion must be separated and noted like a section number 6. Please clarify the lines 678-680.

Author Response

(The authors gave the same response as above.)
